# Dietary Malondialdehyde Damage to the Growth Performance and Digestive Function of Hybrid Grouper (*Epinephelus fuscoguttatus*♀ *× E. lanceolatu*♂)

**DOI:** 10.3390/ani13193145

**Published:** 2023-10-08

**Authors:** Jiongting Fan, Yumeng Zhang, Hang Zhou, Yu Liu, Yixiong Cao, Xiaomei Dou, Xinlangji Fu, Junming Deng, Beiping Tan

**Affiliations:** College of Fisheries, Guangdong Ocean University, Zhanjiang 524088, China; 2112101099@stu.gdou.edu.cn (J.F.); 2112001111@stu.gdou.edu.cn (Y.Z.); wumingshi11@stu.gdou.edu.cn (H.Z.); liuyu@stu.gdou.edu.cn (Y.L.); 2112001050@stu.gdou.edu.cn (Y.C.); 2112001007@stu.gdou.edu.cn (X.D.); 2112001038@stu.gdou.edu.cn (X.F.); tanbp@gdou.edu.cn (B.T.)

**Keywords:** malondialdehyde, hybrid grouper, growth, oxidative stress, intestinal health, Nrf2/Keap1 pathway

## Abstract

**Simple Summary:**

Few studies have been conducted on the harmful effects of MDA levels in aquafeeds on animals. This study aimed to investigate the effect of dietary MDA on the growth performance and digestive function of hybrid groupers (*Epinephelus fuscoguttatus*♀ × *E. lanceolatu*♂). Six isoproteic and isolipidic diets were formulated to contain 0.03, 1.11, 2.21, 4.43, 8.86 and 17.72 mg/kg of MDA, respectively. Each diet was randomly assigned to triplicates of 30 juveniles (14.47 ± 0.01 g) per tank in a flowing water culture system. It shows that the effect of MDA on hybrid groupers showed a dose-dependent effect on this study. A low dose of dietary MDA had limited effects on the growth performance and intestinal health of hybrid group-ers, while a high concentration damaged the gastrointestinal structure and negatively impacted the intestinal digestive and antioxidant functions, and thereby impaired the growth and health of hybrid groupers.

**Abstract:**

Malondialdehyde (MDA) is the dominant component of lipid peroxidation products. Improper storage and transportation can elevate the lipid deterioration MDA content of diets to values that are unsafe for aquatic animals and even hazardous to human health. The study aimed to investigate the effect of dietary MDA on growth performance and digestive function of hybrid grouper (*Epinephelus fuscoguttatus*♀ *× E. lanceolatu*♂). Six isoproteic and isolipidic diets were formulated to contain 0.03, 1.11, 2.21, 4.43, 8.86 and 17.72 mg/kg MDA, respectively. The study shows that the increased dietary MDA content linearly reduced the growth rate, feed utilization, body index and body lipid content of hybrid grouper, while the low dose of dietary MDA (≤2.21 mg/kg) created no difference. Similarly, dietary MDA inclusion linearly depressed the activities of intestinal digestive and absorptive enzymes as well as antioxidant enzymes, enhanced the serum diamine oxidase activity, endotoxin level and intestinal MDA content. A high dose of MDA (≥4.43 mg/kg) generally impaired the gastric and intestinal mucosa, up-regulated the relative expression of Kelch-like ECH-associated protein 1 but down-regulated the relative expression of nuclear factor erythroid 2-related factor 2 in hindgut. In conclusion, the effect of MDA on hybrid grouper showed a dose-dependent effect in this study. A low dose of dietary MDA had limited effects on growth performance and intestinal health of hybrid grouper, while a high concentration damaged the gastrointestinal structure, depressed the intestinal digestive and antioxidant functions, and thereby impaired the growth and health of hybrid grouper.

## 1. Introduction

Recently, the rapid development of aquaculture has increased the demand for aquafeed, and a high-quality diet is the key factor that benefits aquaculture by reducing uneaten food dispersion and organic pollution. As the major energy source for fish, lipids also supply essential fatty acids for the growth and reproduction of fish [1]. However, lipids are susceptible to oxidation under high temperature, high humidity, and oxygen exposure [2,3]. Deteriorated lipids in aquafeed may cause growth retardation and even deformities of fish [4]. However, the main factors underlying the potential harm of oxidized lipids are still not well understood.

Malondialdehyde (MDA) is the dominant component of lipid peroxidation products present in feed ingredients and compound feed [5,6] and thus is often used as an indicator to evaluate the deterioration degree of lipid oxidation [7]. MDA may also crosslink the amino group of protein and phospholipid and reduce the mobility of cells, thereby altering their physiological function and affecting the growth and health of animals [7,8]. Moreover, MDA may damage DNA and mitochondria, leading to massive apoptosis [9,10]. In terrestrial mammals, dietary MDA induced the apoptosis of bone mesenchymal stem cells in mouse [11] and inhibited the mitochondrial respiratory function in mice [10]. The intestine is the main digestive organ of fish and also the main site of exposure to dietary MDA. A previous study has shown that the adding of MDA into the culture medium inhibited the growth of intestinal mucosal cells of grass carp *Ctenopharyngodon idella* in vitro [12]. Further, dietary MDA caused intestinal oxidative stress via the glutathione/glutathione transferase pathway [13] and increased intestinal permeability [6], and thereby impaired the intestinal health of grass carp. However, limited information is available on the influence of dietary MDA on the digestive function of carnivorous fish.

Hybrid grouper (*Epinephelus fuscoguttatus*♀ *× E. polyphekadion*♂), a typical carnivorous fish, has a fast growth rate and strong disease resistance [14]. This species has been reported to be tasty and rich in eicosapentaenoic acid and docosahexaenoic acid [15]. Currently, hybrid grouper is widely farmed in Southeast Asia and the southeast coast of China [16], as are other fish species, for commercial purposes. The dietary lipid requirement of grouper is relatively high, ranging from 9.7% to 14.0% [17], while the dietary lipid requirement of *Trachinotus ovatus* is 10% [18]. Thus, the commercial compound feed for grouper may induce high dietary MDA content during the production, transportation, and storage process. The gastrointestinal tract is the main place for the digesting, absorbing and transporting of nutrients; thus, intestinal health is crucial to hybrid grouper. However, no study was conducted to assess the effects of dietary MDA on the digestive function of hybrid grouper so far. Thus, the objective of this study was to examine the influences of dietary MDA on the growth performance and digestive function of hybrid grouper.

## 2. Materials and Methods

### 2.1. Animal Ethics Statement

This experiment was approved by the Animal Research and Ethics Committee of Guangdong Ocean University (GDOU-IACUC-2021-A2113), and all experimental procedures were conducted under the Guidance of the Care and Use of Laboratory Animals in China (GB/T 35892-2018).

### 2.2. Experimental Feed

Fish meal, soybean meal and soy protein concentrate were used as the major protein sources, while soybean oil and soybean lecithin were used as the lipid sources to formulate a basal diet (Table 1). The feed ingredients (except for lipid sources) are crushed and then passed through a sieve with a diameter of 60-mesh, then weighed according to the feed formula and mixed well. Soybean oil, soy lecithin and a certain amount of water were then added to the compound, mixed adequately and passed through a twin-screw extruder (F-26 type; South China University of Technology, Guangzhou, China) to make 2.5-mm pellet. The feeds were placed in the shade to dry naturally for two days and then stored at −20 °C.

Six experimental diets (M0, M1, M2, M4, M8, and M16) were formulated by splashing different concentrations of MDA solution before each feeding as an addition to different dietary MDA levels. The MDA standard solution (100 μg/mL) was prepared according to GB/T 5009.181-2016: dissolve 0.315 g 1,1,3,3-tetrathoxypropane (purity ≥ 97%; Macklin Biochemical Co., Ltd., Shanghai, China) in distilled water and dilute to 1000 mL, store at 4 °C. Experimental diets were sprayed with 10 mL of MDA solution with different MDA levels (0, 0.22, 0.44, 0.88, 1.76 and 3.52 mL of MDA standard solution was diluted to 10 mL with distilled water, respectively), which referenced the MDA concentration of commercial feeds for grouper and experimental diets for grass carp [6] and hybrid grouper [19]. To determine the MDA content in diets, crude lipid in the diets was firstly extracted and then, using commercial kits (Nanjing Jiancheng Bioengineering Institute, Nanjing, China) for measuring MDA content according to the instruction, the actual MDA content was measured to be 0.03, 1.11, 2.21, 4.43, 8.86 and 17.72 mg/kg of diets M0, M1, M2, M4, M8, and M16, respectively.

### 2.3. Fish and Experimental Conditions

Juvenile hybrid groupers were purchased from a local fish farm (Zhanjiang, China). Of them, 540 healthy juveniles with similar size (average body weight 14.77 ± 0.01 g) were randomly distributed into 18 tanks (300 L) with 30 juveniles per tank (triplicate groups per dietary treatment). Fish were fed twice (8:00 and 17:00) daily to appearance of satiation. During the 8-week feeding period, a flowing water culture system (one-way non-circulating) was used: water flow rate 0.3 m^3^/h, water temperature 26–30 °C, dissolved oxygen ≥ 6 mg/L, pH 7.5–8.0, ammonia nitrogen value 0.05–0.1 mg/L, and salinity 26–28‰.

### 2.4. Sample Collection and Pre-Treatment

All fish were fasted for 24 h before sampling. Fish in each tank were counted and weighed, and then anesthetized with 100 mg/L eugenol solution (Macklin Biochemical Co., Ltd., Shanghai, China). Four fish were randomly selected from each tank, and their body weight, body length, liver weight, and viscera weight were measured and recorded for analysis of body index. Three fish per tank were randomly taken for analysis of body composition.

Blood was taken from the tail vein of five fish per tank; the blood samples stood for 12 h at 4 °C and were then centrifuged at 805× *g* for 10 min. The serum was stored at −80 °C for determination of biochemical parameters. The stomach, foregut (one third of the intestinal segment near the stomach end) and hindgut (one third of the intestinal segment near the excretory opening) of six fish were randomly taken from each tank, quickly placed in liquid nitrogen for temporary storage, and then placed at −80 °C in a refrigerator for analysis of enzyme activities. Another hindgut of four fish per tank was taken into RNAlater (Ambion, Austin, TX, USA), stored at 4 °C for a day and then placed at −80 °C in a refrigerator for analysis of relative mRNA expression.

Additionally, the hindgut of four fish per tank was placed in a 4% paraformaldehyde solution for preparing hematoxylin-eosin (H&E) staining sections, and of four fish per tank of M0, M4, and M16 diet in a 2.5% glutaraldehyde fixation solution for preparing transmission electron microscopy (TEM) sections, respectively. The stomachs of M0, M4, and M16 diet fish were placed in 2.5% glutaraldehyde fixation solution for preparing scanning electron microscopy sections (SEM).

### 2.5. Chemical Composition Analysis

The nutritional composition of experimental diets and fish body were measured using the standard AOAC methods [20]: dry matter dried to a constant weight at 105 °C; crude protein (N × 6.25) by the regular Kjeldahl method (Kjeltec^TM^ 8400; Foss Inc., Hoganas, Sweden); crude lipid by the Soxhlet method with petroleum extraction; ash by incineration at 550 °C for 16 h.

### 2.6. Biochemical Indexes Analyses

The stomach/foregut/hindgut samples (about 0.1 g) plus nine times of saline solution were homogenized in an ice bath, and the supernatant was taken after centrifugation (805× *g* for 10 min) at 4 °C. Total antioxidant capacity (TAC), catalase (CAT) and superoxide dismutase (SOD) activities as well as MDA content in serum and hindgut, and lipase, maltase and amylase activities in foregut, were measured by commercial kits (Nanjing Jiancheng Bioengineering Institute, Nanjing, China) according to the instruction. The activity/content of pepsin in stomach, peroxidase (POD), glutathione peroxidase (GPx), glutathione reductases (GR), diamine oxidase (DAO) and endotoxin in serum and hindgut, and trypsin, pepsin, Na^+^/K^+^-ATPase and Ca^2+^/Mg^2+^-ATPase in foregut were measured using commercial kits (Shanghai Enzyme Linked Biotechnology Co., Ltd., Shanghai, China) according to the instruction.

### 2.7. Histological Observation

The hindgut was soaked in 4% paraformaldehyde solution for 24 h and then transferred to 70%, 80%, 90%, 95% and 100% ethanol concentrations, respectively, and then soaked in xylene solution to make it transparent and embedded in paraffin. The hindgut was then cut into thin sections, stained with H&E, sealed with neutral gum, and the sections were observed using an inverted fluorescence microscope (Nikon Eclipse Ni-U; Nikon, Tokyo, Japan). Then, ten folds per section were randomly selected to measure the fold height, fold width and muscle thickness by image acquisition software (CellSens Standard 1.8; Pooher Optoelectronics (Shanghai) Technology Co., Ltd., Shanghai, China). Hindgut and stomach were placed in 2.5% glutaraldehyde fixed solution for 24 h, and the intestinal transmission electron microscopy sections were prepared with the method described by Huang, et al. [21]. The scanning electron microscopy sections of stomach were prepared with the method of Chen [6].

### 2.8. Extraction of RNA and Real-Time Quantitative PCR Analysis

Total RNA from the hindgut was extracted using TransZol Up Plus RNA Kit (Beijing TransGen Biotech Co., Ltd., Beijing, China) and the integrity of total RNA was validated by agarose gel electrophoresis; the purity and concentration of total RNA were analyzed spectrophotometrically (A260:280 nm). Then, reverse-transcription of RNA to cDNA was performed using PrimeScript^TM^ RT Reagent Kit (Takara Bio Inc., Tokyo, Japan) according to the instruction. Based on the sequences in Gen Bank, specific primers were designed using Primer 5, as shown in Table 2. Real-time quantitative PCR was executed on a quantitative thermal cycler (Light Cycler 480, Roche Diagnostics, Switzerland). The 10 μL reaction volume for RT-qPCR contains 5 μL 2X SYBR^®^ Green Pro Taq HS PremixII (Accurate Biotechnology (Hunan) Co., Ltd., Hunan, China), 3.8 L RNase-free water, 1 μL of cDNA and 0.1 μL of two primers (5 μM each). The thermal profile for real-time quantitative PCR was 95 °C for 30 s, followed by 40 cycles of 95 °C for 5 s, 60 °C for 30 s, and one cycle of 95 °C for 5 s, 60 °C for 60 s, then 50 °C for 30 s. *β-actin* was considered as the reference gene. Gene expression results were calculated according to the 2^−ΔΔCT^ method [22].

### 2.9. Calculation and Statistical Analysis

Formulas for growth performance and body index calculation are as follows:$$\mathrm{Weight}\;\mathrm{gain}\;\mathrm{rate}\;\left(\mathrm{WGR}\right)=100\;\times \left(W_{f}-W_{i}\right)/W_{i};$$
$$\mathrm{Weight}\;\mathrm{gain}\;\mathrm{rate}\;\left(\mathrm{WGR}\right)=100\;\times \left(W_{f}-W_{i}\right)/W_{i};$$
$$\mathrm{Specific}\;\mathrm{growth}\;\mathrm{rate}\;(\mathrm{SGR},\%/d)\;=100\;\times \left(\mathrm{Ln}\;W_{f}-\mathrm{Ln}\;W_{i}\right)/t;$$
$$\mathrm{Feed}\;\mathrm{conversion}\;(\mathrm{FCR})=F/(W_{f}-W_{i})$$
$$\mathrm{Protein}\;\mathrm{efficiency}\;\mathrm{ratio}\;\left(\mathrm{PER}\right)=\left(W_{f}-W_{i}\right)/(F\;\times \;CP);$$
$$\mathrm{Hepatosomatic}\;\mathrm{index}\;(\mathrm{HSI},\;\%)=100\;\times \;W_{l}/W_{b};$$
$$\mathrm{Viscerosomatic}\;\mathrm{index}\;(\mathrm{VSI},\;\%)=100\;\times \;W_{v}/W_{b};$$
$$\mathrm{Condition}\;\mathrm{factor}\;(\mathrm{CF},\;g/{\mathrm{cm}}^{3})=100\;\times \;W_{b}/L^{3};$$
where *W*_f_ and *W*_i_ are the final body weight (g) and initial body weight (g); *t* is feeding days (d); *F* is feed intake (g); *CP* is crude protein content of feed (%); *W*_b_, *L*, *W*_v_ and *W*_1_ are body weight (g), body length (cm), viscera weight (g) and liver weight (g) of fish. Additionally, feed intake is obtained by feeding daily on a full stomach and weighing the weight of feed remaining after feeding.

The data were analyzed by a one-way analysis of variance followed by Duncan’s multiple range tests. Additionally, the orthogonal polynomial contrasts analysis was used to detect the potential linear and quadratic effects of dietary MDA level. Statistical analysis was performed using SPSS 21.0 (SPSS Inc., Chicago, IL, USA) for Windows; the significant differences were set at the level of *p* < 0.05.

## 3. Results

### 3.1. Growth Performance

The survival rate of grouper varied from 90.00% to 96.67%, which did not differ from each other (*p* > 0.05, Table 3). The final body weight, WGR, SGR and PER showed highly significant negative linear (*p* < 0.01) and quadratic (*p* < 0.01) relationships with dietary MDA content; thereinto, the final body weight, WGR and PER were significantly lower in the M8 and M16 groups compared to the M0 and M1 groups, and the SGR was significantly lower in the M16 group compared to the M0 and M1 groups (*p* < 0.05). Conversely, the FCR showed a highly significant positive linear (*p* < 0.01) and quadratic (*p* < 0.01) relationship with dietary MDA level, which was significantly higher in the M16 group compared to the M0, M1 and M2 groups (*p* < 0.05). Additionally, the HSI, VSI and CF showed highly significant negative linear (*p* < 0.01) and quadratic (*p* < 0.01) relationships with dietary MDA level; thereinto, the VSI was significantly lower in the M8 and M16 groups compared to the M0 group (*p* < 0.05), while no significant differences in the HSI and CF were observed among the treatment groups (*p* > 0.05).

### 3.2. Whole Body Composition

Dietary MDA level did not affect the whole-body moisture and crude protein contents (*p* > 0.05, Table 4). The whole-body crude lipid content displayed a highly significant negative linear (*p* < 0.01) and quadratic (*p* < 0.01) relationship with dietary MDA level, which was significantly lower in the M8 and M16 groups compared to the M0, M1 and M2 groups (*p* < 0.05). Conversely, the whole-body crude ash content showed a significant positive linear (*p* = 0.01) and quadratic (*p* < 0.01) relationship with dietary MDA level, which was significantly higher in the M8 and M16 groups compared to the M0 and M1 groups (*p* < 0.05).

### 3.3. Digestion and Absorption Enzymes Activities

The pepsin activity showed a significant negative linear (*p* = 0.01) and quadratic (*p* < 0.01) relationship with dietary MDA level, whereas no significant difference was found among the treatment groups (*p* > 0.05; Table 5). Similarly, the intestinal trypsin, lipase, amylase, maltase and Na^+^/K^+^-ATPase activities exhibited highly significant negative linear (*p* < 0.01) and quadratic (*p* < 0.01) relationships with dietary MDA content, whereas no significant differences (*p* > 0.05) were found in the trypsin activity among the M0, M1 and M2 groups, in the amylase activity between the M0 and M1 groups, in the lipase, maltase and Na^+^/K^+^-ATPase activities among the treatment groups. However, dietary MDA content had no significant effect on the intestinal Ca^2+^/Mg^2+^-ATPase activity (*p* > 0.05).

### 3.4. Intestinal Permeability

The serum endotoxin level was significantly higher in the M4, M8 and M16 groups compared to the M0 and M1 groups (*p* < 0.05). Similarly, Serum diamine oxidase activity including the M4, M8, M16 with M2 groups was significantly higher compared with the M0 group (*p* < 0.05; Figure 1).

### 3.5. Intestine and Stomach Tissue Structure

#### 3.5.1. SEM of Gastric Mucosa Cells

Observation of the surface structure of gastric mucosa (Figure 2) revealed that the gastric mucosal cells of hybrid grouper in the M0 group were normal, with clear cell boundaries and tightly arranged cells; a few cells in the M4 group were broken and the other parts were relatively intact; the gastric mucosal cells in the M16 group were severely damaged.

#### 3.5.2. Intestinal Morphology

Dietary MDA level had no significant effects on intestinal fold width and muscular thickness (*p* > 0.05, Figure 3 and Table 6). However, the fold height displayed a highly significant negative linear (*p* < 0.01) and quadratic (*p* < 0.01) relationship with dietary MDA level although no significant difference was observed among the treatment groups (*p* > 0.05).

#### 3.5.3. TEM of Intestinal Mucosal Cells

The TEM observation of intestinal mucosal cells showed that the M0 group had intact tighter junctional structure and cell structure (Figure 4). However, cells in the M4 and M16 groups showed gaps at the microvilli ends, indicating that the connecting structures between intestinal mucosal cells were damaged.

### 3.6. Antioxidant-Related Index

The serum SOD (*p* = 0.01, *p* = 0.01), CAT (*p* = 0.02, *p* = 0.03), POD (*p* = 0.01, *p* < 0.01), GPx (*p* < 0.01, *p* < 0.01), GR (*p* < 0.01, *p* < 0.01) and TAC (*p* < 0.01, *p* = 0.03) activities exhibited significant or highly significant negative linear and quadratic relationships with dietary MDA content (Table 7), whereas no significant differences (*p* > 0.05) were observed in the SOD, CAT and TAC activities among the treatment groups, in the POD activity among the M0, M1, M2 and M4 groups, or in the GPx and GR activities among the M0, M1 and M2 groups. Conversely, the serum MDA level showed a highly significant positive linear (*p* < 0.01) and quadratic (*p* < 0.01) relationship with dietary MDA level, which was significantly higher in the M2, M4, M8 and M16 groups compared to the M0 group (*p* < 0.05).

Dietary MDA content had no significant effects on the intestinal SOD and POD activities (*p* > 0.05, Table 8). The intestinal CAT (*p* < 0.01, *p* = 0.02), GPx (*p* < 0.01, *p* < 0.01), GR (*p* = 0.03, *p* = 0.01) and TAC (*p* = 0.04, *p* = 0.03) activities exhibited significant or highly significant negative linear and quadratic relationships with dietary MDA content, whereas no significant differences (*p* > 0.05) were found in the CAT and TAC activities among the treatment groups, or in the GPx and GR activities among the M0, M1 and M2 groups. Conversely, the intestinal MDA level showed a significant positive linear (*p* = 0.02) and quadratic (*p* = 0.02) relationship with dietary MDA level, but no significant difference was observed among the treatment groups (*p* > 0.05).

### 3.7. The Relative Expression of Oxidative Stress-Related Factors

The relative expression level of intestinal *keap1* increased with the rising dietary MDA level, which was significantly higher in the M16 group compared to the M0 group (*p* < 0.05; Figure 5). Conversely, the relative expression level of intestinal *nrf2* decreased with the rising dietary MDA content, which was significantly lower in the M8 and M16 groups compared to the M0 group (*p* < 0.05).

## 4. Discussion

MDA is an important indicator of lipid peroxidation and may have a negative impact on the growth and health of animals. Previous research has confirmed that the addition of MDA to the drinking water (0.1–10.0 μg/g/day) of mice increased the mortality rate and damaged the respiratory function of mitochondria in the mice [10]. Similarly, the growth rate (WGR, SGR), feed utilization (FCR, PER), and body lipid content of the hybrid grouper were depressed by dietary supplementation with 8.86 and 17.72 mg/kg MDA in this study. Additionally, previous studies showed that oxidized oil inhabited the growth rate of orange spotted grouper *Epinephelu coioides* [23,24], Japanese sea bass *Lateolabrax japonicus* [25], *Rhynchocypris lagowski* Dybowski [26], Wuchang bream *Megalobrama amblycephala* [27], tilapia *Oreochromis niloticus* [28] and pearl gentian grouper *Epinephelus fuscoguttatus*♀ × *E. lanceolatus*♂ [19]. It deemed that the adverse effect of oxidized oil on the lipid digestion and absorption of grass carp [6] may mainly be attributed to dietary MDA present in oxidized oil. Thus, MDA as the dominant component of lipid peroxidation products may be a major factor underlying the detrimental effects of oxidized oils on fish growth [5,6]. These results indicate that MDA mainly reduces the growth performance of hybrid grouper by decreasing the utilization of lipids and proteins.

Body index is also an important parameter reflecting fish growth, and the viscera and muscle tissue are important contributors to body lipid deposition. The increased body indexes (VSI, HSI and CF) were mainly related to the high dietary lipid content [19]. In this study, the body indexes (VSI, HSI, CF) and body lipid content depressed linearly with the increasing dietary MDA level, although all experimental diets were iso-lipidic. A similar result was reported by Chen et al., who found that a high dose of MDA (61.59–185.04 mg/kg) significantly reduced the body lipid content and HSI of grass carp. These results indicated that dietary MDA might depress the utilization of dietary lipid, but the actual mechanism remained unclear.

The gastrointestinal tract is an important place for nutrient digestion and absorption in fish. Digestive enzymes hydrolyze large molecules of nutrients into small molecules for absorption and utilization by fish [29]. Hence, digestive enzyme activities can reflect the digestive capacity of fish [19]. In this study, dietary MDA inclusion level exceeding 8.86 mg/kg reduced the intestinal digestion (stomach, trypsin, lipase, amylase and maltase) and absorption (Na^+^/K^+^-ATPase, Ca^2+^/Mg^2+^-ATPase) enzyme activities, suggesting that high doses of dietary MDA impaired the digestion and absorption capacity of hybrid grouper. Thus, the depressed intestinal digestion and absorption enzyme activities partly explained the decreased growth performance of hybrid grouper fed with diets M8 and M16. Meanwhile, a previous study also showed that oxidized oil inhibited the intestinal digestive enzymes activities in tilapia, which might be attributed to MDA present in oxidized oil [28].

The digestive capacity of fish is closely related to the development of the intestine, which is a key sign to ensure the growth and health of fish [30]. Specifically, fold height affects the surface area of intestinal absorption, and intestinal muscle thickness is related to intestinal peristalsis [31]. In this study, dietary inclusion of 17.72 mg/kg MDA markedly reduced the intestinal fold height, indicating that high dietary MDA reduced intestinal digestive and absorptive area and thereby inhibited the digestive and absorptive function of hybrid group. Similarly, dietary inclusion of 61.59–185.04 mg/kg MDA damaged the intestinal villi and microvilli and thereby reduced the digestion and absorption capacity of grass carp [6]. Additionally, previous studies also found that oxidized oil had significant effects on the intestinal fold height in grass carp [32], yellow catfish *Pelteobagrus fulvidraco* [33] and tilapia [6,28,33], which may be related to MDA present in oxidized oil.

Intact gastrointestinal mucosa are important for aquatic organisms and help in nutrient absorption. The integrity of intestinal tight junctions is related to the permeability of fish intestine, and the destruction of intestinal mucosa will lead to increased intestinal permeability [34]. Elevated intestinal permeability may increase the invasion of external antigens and harmful microorganisms, leading to a range of intestinal health problems. At the same time, increased intestinal permeability resulted in intestinal DAO and endotoxin being released into the blood [6,35]. A previous study in vitro has shown that MDA at the concentration range of 1.23–9.89 μmol/L damaged the dissociated intestinal epithelial cells from grass carp by dysregulating cell growth, cell morphology, cell structure, and lipid peroxidation [12]. Meanwhile, dietary MDA may also act on the structural points of the cell membrane and disrupt the structural integrity of intestine in grass carp [6]. In this study, dietary MDA inclusion above 4.43 mg/kg increased the serum DAO activity and endotoxin content and impaired gastric mucosa and intestinal epithelial cells. These results indicated that high dietary MDA can directly damage the mucosal structure of the gastrointestinal tract and thereby increase the permeability of intestinal mucosa in hybrid grouper.

MDA is a potential biomarker of toxicity; dietary MDA may promote lipid peroxidation of cell membranes [12]. Thus, it may result in a negative impact on the antioxidant system of fish. The antioxidant system includes the antioxidant enzymes and non-enzymatic antioxidant substances; the antioxidant condition can reflect the health status of fish [36]. Additionally, excessive oxidative production might reduce the activities of antioxidant enzymes and thereby disrupt the balance of oxidation and antioxidation [37,38,39]. A previous study showed that dietary inclusion of 61–185 mg/kg MDA up-regulated the mRNA expression of genes related to the intestinal glutathione/glutathione transferase pathway in grass carp [13]. The Nrf2/Keap1 pathway is an important pathway to regulate the oxidative stress in the body [30]; it might improve the intestinal antioxidant status by inhibiting *keap1* and promoting the nuclear translocation of *nrf2*, and even regulate the activities of antioxidant-related enzymes [40]. In this study, dietary MDA inclusion levels exceeding 8.86 mg/kg markedly down-regulated the relative expression of *nrf2* but up-regulated the relative expression of *keap1*, accompanied by the increased intestinal MDA content and decreased intestinal antioxidant enzymes activities. These results indicated that a high dose of dietary MDA might cause intestinal oxidative stress by inhibiting the Nrf2/Keap1 pathway.

## 5. Conclusions

In conclusion, dietary MDA levels up to 2.21 mg/kg showed no significant negative effects on growth performance, feed utilization or gastrointestinal structure and function of hybrid grouper. However, further inclusion (≥4.43 mg/kg) of dietary MDA markedly impaired the structure of gastrointestinal mucosa, increased the intestinal permeability, inhibited the Nrf2/Keap1 pathway and induced oxidative stress, depressed the intestinal antioxidant status and even digestive enzyme activities, and thereby inhibited the growth performance of hybrid grouper. Thus, the potential hazard of dietary MDA on aquatic animals must be paid enough attention.

## Figures and Tables

**Figure 1 animals-13-03145-f001:**
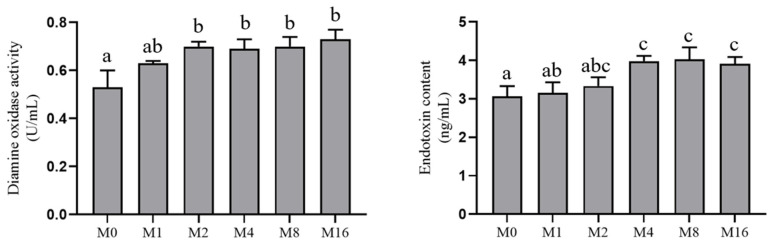
Effect of dietary malondialdehyde level on intestinal permeability-related parameters in serum of hybrid grouper. Bars represent serum diamine oxidase activity (**left**) and endotoxin levels (**right**), respectively. M0, M1, M2, M4, M8 and M16 represent dietary groups sprayed with 0, 1, 2, 4, 8 and 16 mg/kg malondialdehyde, respectively. Values are means with standard errors represented by vertical bars (*n* = 3). ^a,b,c^ Means without a common superscript differ significantly (*p* < 0.05).

**Figure 2 animals-13-03145-f002:**
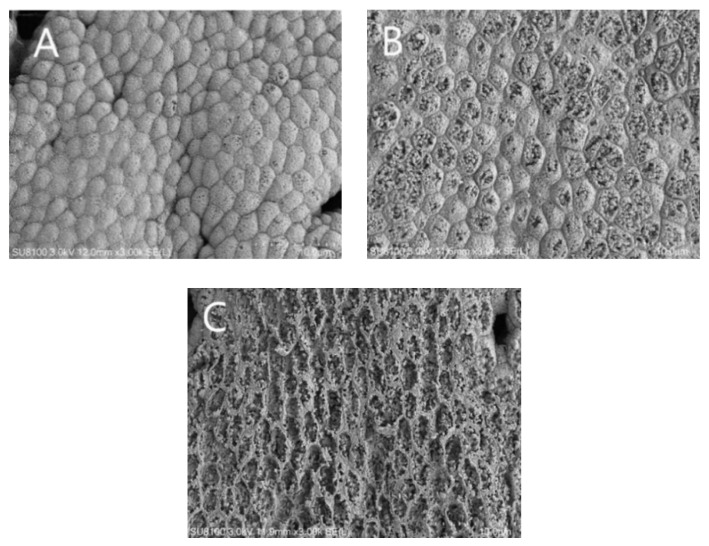
Scanning electron microscopy of gastric mucosa in hybrid grouper fed with various dietary malondialdehyde levels. (**A**), M0 group sprayed with 0 mg/kg malondialdehyde; (**B**), M4 group sprayed with 4 mg/kg malondialdehyde; (**C**), M16 group sprayed with 16 mg/kg malondialdehyde. The magnification was ×3000 and the minimum scale (lower right) 10.0 μm. The gastric mucosal cells in the M0 group were normal, with clear cell borders and tightly arranged cells; a small number of cells in the M4 group were broken and the rest were relatively intact; the gastric mucosal cells in the M16 group were severely damaged.

**Figure 3 animals-13-03145-f003:**
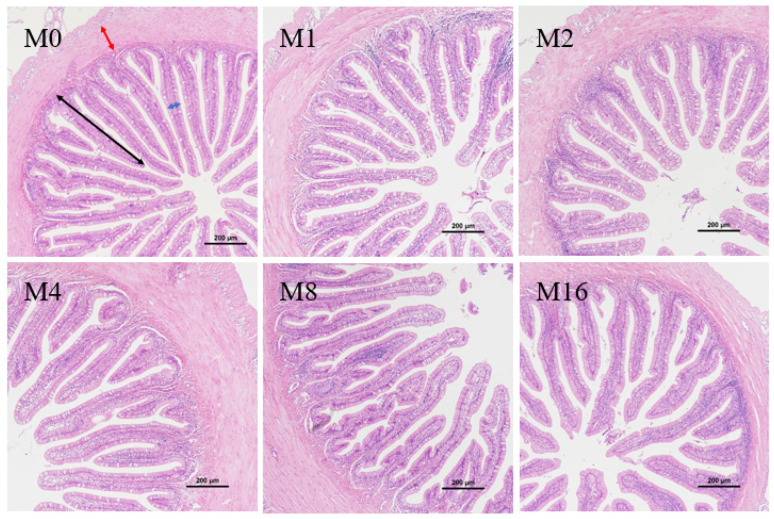
Effect of dietary malondialdehyde level on the hindgut histomorphology of hybrid grouper fed with various dietary malondialdehyde levels. M0, M1, M2, M4, M8 and M16 represent dietary groups sprayed with 0, 1, 2, 4, 8 and 16 mg/kg malondialdehyde, respectively. Black arrows: fold height (μm); blue arrows: fold width (μm); red arrows: muscular thickness (μm). The magnification was ×100 and the minimum scale (lower right) 200 μm.

**Figure 4 animals-13-03145-f004:**
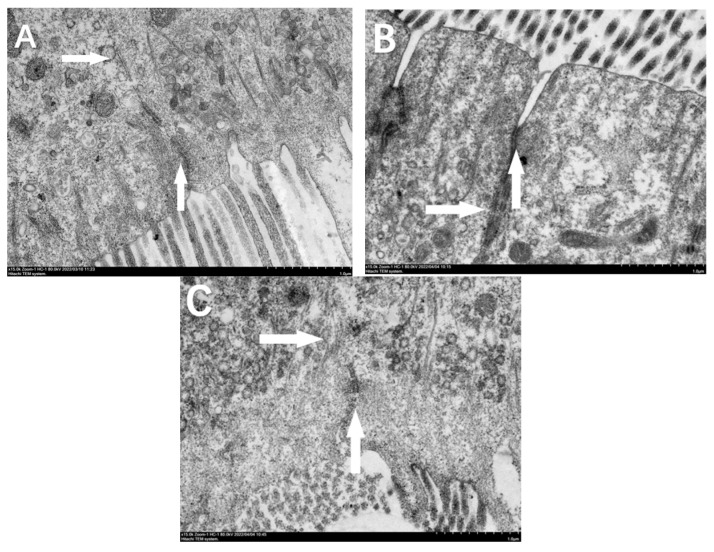
Transmission electron microscopy of intestinal mucosa in hybrid grouper fed with various dietary malondialdehyde levels. (**A**), M0 group sprayed with 0 mg/kg malondialdehyde; (**B**), M4 group sprayed with 4 mg/kg malondialdehyde; (**C**), M16 group sprayed with 16 mg/kg malondialdehyde. Vertical arrows indicate the junctions of microvilli ends; horizontal arrows indicate the cell membrane structure. The magnification was ×15,000 and the minimum scale (lower right) 1.0 μm. The junctions of microvilli ends in the M0 group were relatively tight and the cell morphology was normal; cells in the M4 and M16 groups showed gaps at the end of microvilli, and the cell membrane of the M16 group was broken.

**Figure 5 animals-13-03145-f005:**
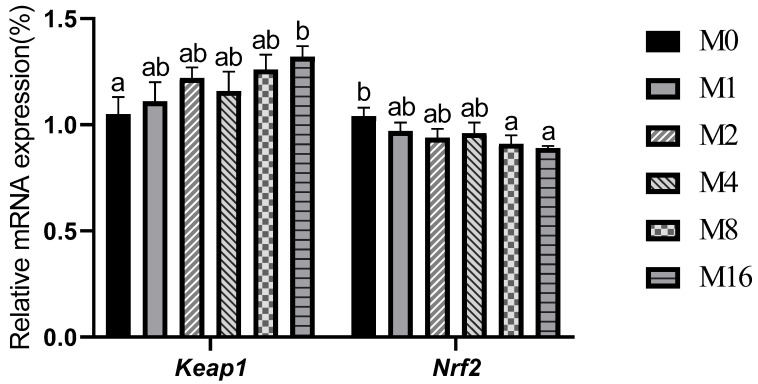
Effect of dietary malondialdehyde level on the relative mRNA expression of oxidative stress-related factors in hindgut of hybrid grouper fed with various dietary malondialdehyde levels. M0, M1, M2, M4, M8 and M16 represent dietary groups sprayed with 0, 1, 2, 4, 8 and 16 mg/kg malondialdehyde, respectively. *Keap1*, Kelch-like ECH-associated protein 1; *Nrf2*, nuclear factor erythroid 2-related factor 2. Values are means with standard errors represented by vertical bars (*n* = 3). ^a,b^ Means with different letters were significantly different (*p* < 0.05).

**Table 1 animals-13-03145-t001:** Ingredients and nutritional composition of the basal experimental diets (% dry matter).

Ingredients	%
Fish meal	45.00
Soybean protein concentrate	21.00
Soybean meal	6.00
Wheat flour	17.91
Soybean oil	5.30
Soybean lecithin	2.00
Ca(H_2_PO_4_)_2_	1.20
Choline chloride	0.40
Vitamin C	0.03
Compound premix ^1^	1.00
Cellulose microcrystalline	0.16
Proximate composition	
Dry matter (DM, %)	92.34
Crude protein (% DM)	49.62
Crude lipid (% DM)	11.07
Ash (% DM)	16.95

^1^ Compound premix was supplied by Beijing Enhalor Biotechnology Co., Ltd., Beijing, China (g kg^−1^ mixture): vitamin A, 500,000 IU; vitamin D_3_, 100,000 IU; vitamin E, 4.00 g; vitamin K_3_, 1.00 g; vitamin B_1_, 0.50 g, vitamin B_2_, 1.00 g; vitamin B_6_, 1.00 g; vitamin B_12_, 0.002 g; nicotinic acid, 4.00 g; calcium pantothenate, 2.00 g; biotin, 0.01 g; folic acid, 0.10 mg; vitamin C, 15.00 g; ferrum, 10.00 g; cuprum, 0.30 g; zinc, 5.00 g; manganese, 1.20 g; iodine, 0.08 g; cobalt, 0.02 g; selenium, 0.03 g.

**Table 2 animals-13-03145-t002:** Primer pair sequences used in real-time PCR.

Target Gene	Primer Sequence	Genbank Accession No.	TM (°C)
*keap1* ^1^	F-TCCACAAACCCACCAAAGTAA R-TCCACCAACAGCGTAGAAAAG	XM_018665037.1	57.62 58.51
*nrf2* ^2^	F-TATGGAGATGGGTCCTTTGGTG R-GCTTCTTTTCCTGCGTCTGTTG	KU892416.1	59.49 60.60
*β-Actin*	F-GGCTACTCCTTCACCACCACA R-TCTGGGCAACGGAACCTCT	AY510710.2	61.72 60.84

^1^ *Keap1*, Kelch-like ECH-associated protein 1. ^2^ *Nrf2*, nuclear factor erythroid 2-related factor 2.

**Table 3 animals-13-03145-t003:** Effect of dietary malondialdehyde level on growth performance, feed utilization and body index of hybrid grouper.

	Diets ^1^						*Pr* > *F* ^2^		
	M0	M1	M2	M4	M8	M16	ANOVA	Linear	Quadratic
Initial body weight (g)	14.77 ± 0.01	14.78 ± 0.01	14.76 ± 0.01	14.77 ± 0.01	14.77 ± 0.01	14.79 ± 0.01	0.43	0.19	0.30
Final body weight (g)	116.39 ± 0.25 ^b^	116.09 ± 1.39 ^b^	113.45 ± 1.59 ^ab^	110.45 ± 0.56 ^ab^	109.21 ± 1.06 ^a^	107.19 ± 4.77 ^a^	0.03	<0.01	<0.01
Weight gain rate (%)	688.21 ± 2.17 ^b^	685.19 ± 9.06 ^b^	668.80 ± 11.80 ^ab^	647.98 ± 4.31 ^ab^	639.23 ± 5.99 ^a^	624.87 ± 33.15 ^a^	0.04	<0.01	<0.01
Specific growth ratio (%/d)	3.62 ± 0.01 ^b^	3.62 ± 0.02 ^b^	3.59 ± 0.02 ^ab^	3.53 ± 0.01 ^ab^	3.51 ± 0.01 ^ab^	3.47 ± 0.08 ^a^	0.03	<0.01	<0.01
Feed conversion ratio	0.83 ± 0.01 ^a^	0.83 ± 0.01 ^a^	0.84 ± 0.01 ^a^	0.87 ± 0.01 ^ab^	0.90 ± 0.02 ^ab^	0.92 ± 0.05 ^b^	0.10	<0.01	<0.01
Protein efficiency ratio	2.44 ± 0.03 ^c^	2.43 ± 0.10 ^c^	2.38 ± 0.03 ^bc^	2.24 ± 0.02 ^abc^	2.16 ± 0.04 ^ab^	2.01 ± 0.05 ^a^	0.02	<0.01	<0.01
Survival rate (%)	96.67 ± 0.00	96.67 ± 3.33	93.33 ± 1.93	93.33 ± 1.93	93.33 ± 3.33	90.00 ± 3.33	0.50	0.05	0.15
Hepatosomatic index (%)	4.55 ± 0.12	4.08 ± 0.09	4.10 ± 0.16	3.98 ± 0.09	3.90 ± 0.27	3.92 ± 0.16	0.06	<0.01	<0.01
Viscerosomatic index (%)	12.79 ± 0.35 ^b^	12.01 ± 0.44 ^ab^	11.86 ± 0.23 ^ab^	11.77 ± 0.28 ^ab^	11.50 ± 0.24 ^a^	11.17 ± 0.39 ^a^	0.02	<0.01	<0.01
Condition factor (g/cm^3^)	3.01 ± 0.09	2.88 ± 0.10	2.81 ± 0.11	2.77 ± 0.07	2.66 ± 0.05	2.70 ± 0.03	0.08	<0.01	<0.01

^1^ M0, M1, M2, M4, M8 and M16 represent diets sprayed with 0, 1, 2, 4, 8 and 16 mg/kg malondialdehyde, respectively. ^2^ Significance probability associated with the F-statistic. ^a,b,c^ Values are means ± standard error (SE) of three replications (*n* = 3). Within a row, means without a common superscript differ significantly (*p* < 0.05).

**Table 4 animals-13-03145-t004:** Effect of dietary malondialdehyde level on the whole-body compositions of hybrid grouper.

	Diets ^1^						*Pr* > *F* ^2^		
	M0	M1	M2	M4	M8	M16	ANOVA	Linear	Quadratic
Moisture (%)	72.01 ± 0.50	71.75 ± 0.24	72.33 ± 0.67	71.68 ± 0.30	71.99 ± 0.28	71.31 ± 0.62	0.74	0.23	0.45
Crude protein (%)	16.14 ± 0.33	16.09 ± 0.75	16.57 ± 0.67	16.58 ± 0.20	16.20 ± 0.67	16.62 ± 0.49	0.96	0.59	0.87
Crude lipid (%)	7.68 ± 0.17 ^c^	7.59 ± 0.04 ^c^	7.17 ± 0.21 ^bc^	6.96 ± 0.15 ^ab^	6.62 ± 0.12 ^a^	6.60 ± 0.23 ^a^	<0.01	<0.01	<0.01
Crude ash (%)	4.50 ± 0.01 ^a^	4.60 ± 0.03 ^ab^	4.72 ± 0.09 ^bc^	4.75 ± 0.00 ^bc^	4.84 ± 0.06 ^c^	4.84 ± 0.06 ^c^	0.03	0.01	<0.01

^1^ M0, M1, M2, M4, M8 and M16 represent diets sprayed with 0, 1, 2, 4, 8 and 16 mg/kg malondialdehyde, respectively. ^2^ Significance probability associated with the F-statistic. ^a,b,c^ Values are means ± standard error (SE) of three replications (*n* = 3). Within a row, means without a common superscript differ significantly (*p* < 0.05).

**Table 5 animals-13-03145-t005:** Effect of dietary malondialdehyde level on the digestion and absorption enzyme activities in the gastrointestinal tract of hybrid grouper.

	Diets ^1^						*Pr* > *F* ^2^		
	M0	M1	M2	M4	M8	M16	ANOVA	Linear	Quadratic
Stomach									
Pepsin (U/mg protein)	56.05 ± 2.82	54.92 ± 3.46	48.68 ± 3.46	46.50 ± 4.07	42.14 ± 2.97	42.42 ± 3.78	0.09	0.01	<0.01
Intestine									
Trypsin (U/μg protein)	0.25 ± 0.02 ^b^	0.23 ± 0.02 ^ab^	0.22 ± 0.02 ^ab^	0.18 ± 0.01 ^a^	0.17 ± 0.02 ^a^	0.18 ± 0.01 ^a^	0.04	<0.01	<0.01
Lipase (U/g protein)	1.65 ± 0.04	1.63 ± 0.04	1.54 ± 0.06	1.53 ± 0.05	1.35 ± 0.04	1.31 ± 0.02	0.09	<0.01	<0.01
Amylase (U/mg protein)	0.38 ± 0.00 ^c^	0.34 ± 0.01 ^bc^	0.32 ± 0.00 ^ab^	0.29 ± 0.03 ^ab^	0.27 ± 0.01 ^a^	0.27 ± 0.02 ^a^	0.01	<0.01	<0.01
Maltase (U/mg protein)	13.93 ± 1.80	13.63 ± 0.94	13.35 ± 0.51	13.05 ± 0.58	11.11 ± 0.73	10.54 ± 0.74	0.08	<0.01	<0.01
Na^+^/K^+^-ATPase (U/mg protein)	1.64 ± 0.12	1.63 ± 0.10	1.56 ± 0.13	1.50 ± 0.11	1.36 ± 0.07	1.27 ± 0.02	0.12	<0.01	<0.01
Ca^2+^/Mg^2+^-ATPase (U/mg protein)	5.71 ± 0.49	5.59 ± 0.71	5.23 ± 0.37	4.98 ± 0.80	4.55 ± 0.89	4.65 ± 0.33	0.74	0.15	0.21

^1^ M0, M1, M2, M4, M8 and M16 represent diets sprayed with 0, 1, 2, 4, 8 and 16 mg/kg malondialdehyde, respectively. ^2^ Significance probability associated with the F-statistic. ^a,b,c^ Values are means ± standard error (SE) of three replications (*n* = 3). Within a row, means without a common superscript differ significantly (*p* < 0.05).

**Table 6 animals-13-03145-t006:** Effect of dietary malondialdehyde level on the intestinal histomorphology of hybrid grouper.

	Diets ^1^						*Pr* > *F* ^2^		
	M0	M1	M2	M4	M8	M16	ANOVA	Linear	Quadratic
Fold height (μm)	643.16 ± 19.54	637.28 ± 19.31	627.47 ± 20.65	607.28 ± 25.21	577.45 ± 16.48	567.77 ± 18.38	0.07	<0.01	<0.01
Fold width (μm)	82.28 ± 2.41	83.02 ± 6.26	79.44 ± 3.85	78.21 ± 3.03	76.33 ± 2.97	77.56 ± 1.86	0.62	0.18	0.18
Muscular thickness (μm)	147.27 ± 7.00	144.53 ± 6.43	141.22 ± 6.18	139.14 ± 8.91	135.67 ± 5.41	135.21 ± 7.69	0.74	0.15	0.25

^1^ M0, M1, M2, M4, M8 and M16 represent diets sprayed with 0, 1, 2, 4, 8 and 16 mg/kg malondialdehyde, respectively. ^2^ Significance probability associated with the F-statistic.

**Table 7 animals-13-03145-t007:** Effect of dietary malondialdehyde level on the antioxidant-related index in serum of hybrid grouper.

	Diets ^1^						*Pr* > *F* ^2^		
	M0	M1	M2	M4	M8	M16	ANOVA	Linear	Quadratic
Superoxide dismutase (U/mL)	62.84 ± 2.26	58.31 ± 5.41	56.84 ± 2.74	56.99 ± 3.69	51.20 ± 3.28	51.44 ± 0.60	0.12	0.01	0.01
Catalase (U/mL)	9.71 ± 1.36	8.13 ± 0.69	7.28 ± 0.40	7.40 ± 1.11	5.75 ± 0.98	5.87 ± 1.03	0.17	0.02	0.03
Peroxidase (U/mL)	0.34 ± 0.01 ^b^	0.34 ± 0.01 ^b^	0.29 ± 0.03 ^ab^	0.27 ± 0.02 ^ab^	0.24 ± 0.01 ^a^	0.26 ± 0.03 ^a^	0.04	0.01	<0.01
Glutathione peroxidase (U/mL)	0.17 ± 0.01 ^c^	0.16 ± 0.01 ^bc^	0.15 ± 0.01 ^bc^	0.14 ± 0.01 ^ab^	0.12 ± 0.01 ^a^	0.12 ± 0.01 ^a^	<0.01	<0.01	<0.01
Glutathione reductase (U/mL)	0.15 ± 0.01 ^c^	0.13 ± 0.01 ^bc^	0.14 ± 0.01 ^bc^	0.11 ± 0.01 ^ab^	0.10 ± 0.01 ^a^	0.09 ± 0.00 ^a^	<0.01	<0.01	<0.01
Total antioxidant capacity (nmol/L)	1.83 ± 0.05	1.80 ± 0.02	1.78 ± 0.03	1.78 ± 0.01	1.74 ± 0.02	1.72 ± 0.04	0.18	<0.01	0.03
Malondialdehyde (nmol/L)	3.30 ± 0.43 ^a^	5.10 ± 1.17 ^ab^	5.99 ± 0.54 ^b^	6.41 ± 1.03 ^bc^	5.96 ± 0.33 ^b^	8.16 ± 0.73 ^c^	<0.01	<0.01	<0.01

^1^ M0, M1, M2, M4, M8 and M16 represent diets sprayed with 0, 1, 2, 4, 8 and 16 mg/kg malondialdehyde, respectively. ^2^ Significance probability associated with the F-statistic. ^a,b,c^ Values are means ± standard error (SE) of three replications (n = 3). Within a row, means without a common superscript differ significantly (*p* < 0.05).

**Table 8 animals-13-03145-t008:** Effect of dietary malondialdehyde level on the antioxidant-related index in intestine of hybrid grouper.

	Diets ^1^						*Pr* > *F* ^2^		
	M0	M1	M2	M4	M8	M16	ANOVA	Linear	Quadratic
Superoxide dismutase (U/μg protein)	0.60 ± 0.01	0.59 ± 0.02	0.58 ± 0.01	0.56 ± 0.03	0.56 ± 0.01	0.55 ± 0.02	0.56	0.11	0.15
Catalase (U/mg protein)	17.46 ± 0.91	16.89 ± 1.15	16.45 ± 0.67	16.05 ± 1.07	14.81 ± 1.34	13.97 ± 0.30	0.17	<0.01	0.02
Peroxidase (U/g protein)	27.13 ± 3.02	25.44 ± 2.33	23.35 ± 3.41	23.39 ± 3.70	21.63 ± 3.98	21.00 ± 3.10	0.79	0.17	0.30
Glutathione peroxidase (U/mg protein)	15.27 ± 1.31 ^b^	14.85 ± 0.89 ^b^	12.17 ± 0.85 ^ab^	10.57 ± 1.47 ^a^	9.78 ± 1.19 ^a^	9.09 ± 0.83 ^a^	0.03	<0.01	<0.01
Glutathione reductase (U/mg protein)	2.49 ± 0.21 ^b^	1.98 ± 0.04 ^ab^	1.93 ± 0.30 ^ab^	1.66 ± 0.07 ^a^	1.64 ± 0.14 ^a^	1.61 ± 0.08 ^a^	0.05	0.03	0.01
Total antioxidant capacity (mmol/g protein)	15.00 ± 2.64	10.70 ± 1.36	10.83 ± 1.22	10.00 ± 2.86	7.17 ± 2.13	8.11 ± 1.30	0.16	0.04	0.03
Malondialdehyde (nmol/g protein)	22.87 ± 5.92	30.77 ± 3.88	34.16 ± 5.41	37.84 ± 5.20	41.93 ± 4.60	43.11 ± 5.32	0.16	0.02	0.02

^1^ M0, M1, M2, M4, M8 and M16 represent diets sprayed with 0, 1, 2, 4, 8 and 16 mg/kg malondialdehyde, respectively.^2^ Significance probability associated with the F-statistic. ^a,b^ Values are means ± standard error (SE) of three replications (n = 3). Within a row, means without a common superscript differ significantly (*p* < 0.05).

## Data Availability

The data that support the findings of this study are available on request from the corresponding author.

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
