# Peer review of "Dietary Malondialdehyde Damage to the Growth Performance and Digestive Function of Hybrid Grouper (Epinephelus fuscoguttatus× E. lanceolatu♂)"

_animals, 2023, doi:10.3390/ani13193145_

Round 1

Reviewer 1 Report

Dear editor,

Greeting!

Review of the manuscript: Dietary malondialdehyde damage the growth per-formance and digestive function of hybrid grouper (Epinephelus fuscoguttatusâ™€× E. lanceolatu♂) . Due to the high fat content in the basic diet of grouper, fat is easy to oxidize and produce malondialdehyde, which will affect animal health. Therefore, the effects of dietary malondialdehyde on growth performance and digestive function of hybrid grouper were investigated in this study. This is an interesting and data-rich study, I think it should to be accepted, but there are still some questions that need to be addressed.

1. In general, how about the concentration of MAD in commercial feed? How to select and do the experimental design concentration in this trial?

2. Line 107, Please provide the specific ingredient content information in the compound premix. And I wonder why vitamin C is listed separately in the feed formula.

3. Line 101, the crude protein, crude fat, crude fiber, water and ash content should be shown.

4. Line 113, I'm curious, if the base diet is sprayed with 10 mL of MDA at different concentrations, how many grams of the diet is used each time, and is 10 mL appropriate?

5. Line 137, Please provide water ammonia nitrogen content.

6. Line 168, In chemical composition analysis, please cite relevant references to support your method.

7.  In Table 2, provide TM (â—¦C) and E-Values (%).

8. Line 322, Formatting errors should be fixed.

9. In the notes in Figure 1, explain exactly what the two figures represent.

10. In 3.5.1. SEM of gastric mucosa cells and 3.5.3. TEM of intestinal mucosal cells, Why only 3 groups were selected for electron microscopy? Please explain the reasons for choosing these groups. The scale should be added to each graph.

11.  In Figure 5, there are spelling errors and unit errors in the ordinate, please check.

Dear editor,

Greeting!

Review of the manuscript: Dietary malondialdehyde damage the growth per-formance and digestive function of hybrid grouper (Epinephelus fuscoguttatusâ™€× E. lanceolatu♂) . Due to the high fat content in the basic diet of grouper, fat is easy to oxidize and produce malondialdehyde, which will affect animal health. Therefore, the effects of dietary malondialdehyde on growth performance and digestive function of hybrid grouper were investigated in this study. This is an interesting and data-rich study, I think it should to be accepted, but there are still some questions that need to be addressed.

1. In general, how about the concentration of MAD in commercial feed? How to select and do the experimental design concentration in this trial?

2. Line 107, Please provide the specific ingredient content information in the compound premix. And I wonder why vitamin C is listed separately in the feed formula.

3. Line 101, the crude protein, crude fat, crude fiber, water and ash content should be shown.

4. Line 113, I'm curious, if the base diet is sprayed with 10 mL of MDA at different concentrations, how many grams of the diet is used each time, and is 10 mL appropriate?

5. Line 137, Please provide water ammonia nitrogen content.

6. Line 168, In chemical composition analysis, please cite relevant references to support your method.

7.  In Table 2, provide TM (â—¦C) and E-Values (%).

8. Line 322, Formatting errors should be fixed.

9. In the notes in Figure 1, explain exactly what the two figures represent.

10. In 3.5.1. SEM of gastric mucosa cells and 3.5.3. TEM of intestinal mucosal cells, Why only 3 groups were selected for electron microscopy? Please explain the reasons for choosing these groups. The scale should be added to each graph.

11.  In Figure 5, there are spelling errors and unit errors in the ordinate, please check.

Reviewer 2 Report

Malondialdehyde (MDA) is the product of lipid peroxidation, can present in feed ingredients and compound feed and affect the growth and health of animals. The present study investigated the effects of dietary MDA on the growth and digestive function of hybrid grouper is appropriate for the journal, However, the manuscript (MS) has some weaknesses and need to be improved.

1.     There are language and technical mistakes in the presentation of the MS. Some statements are too redundant. For example, in the sentence (lines 111-113), “add different dietary MDA levels” with “different concentrations of MDA solution” has similar meaning.

2.     There is insufficient information given about the study design and analysis in M&Ms.

1)     In 2.2 section, there only had the MDA content in experimental diets, but the analysis way of the DMA content in diet is absent.

2)     The experimental diets containing different level of MDA were made by splashing MDA solution. In this process, manual splashing or automatic splashing? How to maintain the consistency of MDA content in each experimental diet?

3)     Author only analyzed the stomach and hindgut structure of fish in M0, M4 and M16 groups by SEM. So, author should state which fishes were collected for preparing SEM in 2.4 section.

4)     The sentence in lines 164-166 showed that only stomach was collected for preparing SEM. How about hindgut?

5)     In 2.7 section, SEM sections of stomach were prepared (lines 208-210). How about hindgut?

Chen, Ye, Cai, Wu, Huang, Wu, Lin, Luo, Zhang, Xiao and Zhou in line 209-210 should be changed to Chen et al.

6)     Generally, the full name of SGR is specific growth rate.

3.     In Results, some data in Tables 4, 5, 7 and 8, were incomplete. Please check and revise them.

4.     In 3.4 section, please check the sentence in lines 322. I can’t understand it.

5.     (2016) in line 478 should be deleted.

6.     In Discussion, author focused on a simple list of similar research results with their results. The analysis between their results and others was insufficient.

Reviewer 3 Report

The present study by Fan and colleagues evaluated the impact of survival growth and morpho- and physiological traits of different concentrations of MDA on the diet of a hybrid species. The paper is interesting and well written, the results are clear and I have no doubt that could be a potential good contribution to the topic. I have some minor comments for the authors that should taken into account for improving the clarity of their paper to general readers. In addition, some methodological information is missing and I ask authors to provide them.

Abstracts:

What are the motivations that pushed researchers to add MDA to diet? In what other areas and purposes is MDA used? Which products contain it? To allow the readers to better understand the article, I believe a sentence is necessary to introduce the topic.

A brief description of the investigated diet treatments is missing. Before describing the results (from line 15), the authors must include a brief note on the doses used.

Finally, given the negative impact of diets containing high additional doses of MDA, the reader wonders why you studied it. As suggested in the first point, it is necessary to motivate your research question by reporting a brief introduction and whether this practice, i.e. adding MDA in the aquafeed, is generally used in aquaculture.

Introduction

Lines 36-37: The authors should justify why new aquafeeds are necessary (i.e., reduction of uneaten food dispersion and organic pollution).

Lines 72-74: It would be interesting to report a comparison of the percentages of fat necessary for the correct development of other fish species of commercial interest. This information will underline the importance of developing correct diet regimes for these species which require a greater intake for correct development and a better welfare state, thus increasing the quality of product.

Lines 71-72: Is the studied species commonly used for commercial purposes? Specify whether this sentence is generalized to different hybrid species, or whether the hybrid species E. fuscoguttatus and E. polyphekadion are bred for commercial purposes. This is an important point to define to provide greater application relevance to the study.

Lines 74-76: The transport process is a very stressful event for the animals, according to my personal experience, which is usually fasting during the process. I would like to ask the authors if this practice, i.e. feeding fish during transportation, is common.

Methods:

Lines 132: How the number of different treatments was randomly distributed among tanks? For example, 18 tanks so, do authors have used 3 per diet regime condition? Please specify this information.

In addition, have you measured the amount of food uneaten, or have fish consumed it every feeding event? Differences in the amount of dispersal uneaten food might be interesting for improving the feeding regime and its composition. Moreover, a brief description of feed intake measurement is necessary (missing in section 2.9).

Statistics and results

In the linear trend, the authors assessed the same relative distance among treatments. However, keep attention that the concentrations do not follow a linear trend (more reasonable to use a geometrical trend). Anyway, the results should be the same but more robust and reliable.  From what my concern is the linear trend analysis does not require posthoc tests because the different groups are considered as covariates and not levels of a factor variable.

Table 1:  I would suggest highlighting the final measure used to assess the difference among diets as reported in the main test. It will allow readers to understand the results better.

The English is fine and only minor edits are necessary

Round 2

Reviewer 2 Report

The manuscript has been improved in this version, but there are some minor mistakes. Such as: ratio in line 261 should be deleted; what’s the meaning of the sentences in lines 344-345.
